# Effect of Temperature, Pressure, and Chemical Composition on the Electrical Conductivity of Schist: Implications for Electrical Structures under the Tibetan Plateau

**DOI:** 10.3390/ma12060961

**Published:** 2019-03-22

**Authors:** Wenqing Sun, Lidong Dai, Heping Li, Haiying Hu, Changcai Liu, Mengqi Wang

**Affiliations:** 1Key Laboratory of High-Temperature and High-Pressure Study of the Earth’s Interior, Institute of Geochemistry, Chinese Academy of Sciences, Guiyang 550081, China; sunwenqing@mail.gyig.ac.cn (W.S.); hepingli_2007@hotmail.com (H.L.); huhaiying@mail.gyig.ac.cn (H.H.); liuchangcai@mail.gyig.ac.cn (C.L.); wangmengqi@mail.gyig.ac.cn (M.W.); 2University of Chinese Academy of Sciences, Beijing 100049, China

**Keywords:** schist, electrical conductivity, high pressure, chemical compositions, conduction mechanism, Tibetan Plateau

## Abstract

The experimental study on the electrical conductivities of schists with various contents of alkali ions (*C*_A_ = K_2_O + Na_2_O = 3.94, 5.17, and 5.78 wt.%) were performed at high temperatures (623–1073 K) and high pressures (0.5–2.5 GPa). Experimental results indicated that the conductivities of schist markedly increased with the rise of temperature. Pressure influence on the conductivities of schist was extremely weak at the entire range of experimental temperatures. Alkali ion content has a significant influence on the conductivities of the schist samples in a lower temperature range (623–773 K), and the influence gradually decreases with increasing temperature in a higher temperature range (823–1073 K). In addition, the activation enthalpies for the conductivities of three schist samples were fitted as being 44.16–61.44 kJ/mol. Based on the activation enthalpies and previous studies, impurity alkaline ions (K^+^ and Na^+^) were proposed as the charge carriers of schist. Furthermore, electrical conductivities of schist (10^−3.5^–10^−1.5^ S/m) were lower than those of high-conductivity layers under the Tibetan Plateau (10^−1^–10^0^ S/m). It was implied that the presence of schist cannot cause the high-conductivity anomalies in the middle to lower crust beneath the Tibetan Plateau.

## 1. Introduction

Electrical conductivity of geological material combined with magnetotelluric (MT) data is very significant in exploring material compositions and temperature–pressure conditions in the Earth’s interior. For most dominant minerals and rocks of the Earth’s crust, upper mantle, and transition zone, the electrical properties have been researched in detail [1,2,3,4,5,6,7,8,9,10,11]. As a significant regional metamorphic zone, the Tibetan Plateau was formed by the subduction of the Indian plate beneath the Eurasian continent. A large amount of metamorphic rocks was widely distributed in the surface of Tibetan Plateau [12,13,14], recording the past geological processes at different depths of the subduction zone. However, the internal state under the Tibetan Plateau has not been very clear until now. Electrical conductivity, an important physical parameter, can be applied to reveal the material compositions and thermodynamic state beneath the Tibetan Plateau. In order to systematically explore the electrical structure under the regional metamorphic belt, electrical properties of the most dominant metamorphic rocks under regional metamorphic belts (gneiss, granulite, and eclogite) were investigated in depth [15,16,17,18]. However, the electrical conductivities of natural schist, a significant medium-grade metamorphic rock, have not been explored in detail to date.

Schist with various chemical compositions can be formed by the metamorphism of pelite, sandstone, and intermediate-acid igneous rock below the depth of the middle crust. During subduction, abundant schist can enter into the deep Earth and be changed into gneiss and granulite after further metamorphism. Schist is widely distributed in the Tibetan Plateau and coexists with slate, gneiss, granulite, among others [13,14]. Therefore, geophysicists should comprehensively research the conductivities of the important metamorphic rocks (schist, gneiss, granulite, and eclogite) and invert the magnetotelluric (MT) data under the Tibetan Plateau. Fuji-ta et al. [16] studied the electrical properties of gneiss at the lower-crust thermodynamic conditions. Effect of temperature and structure on gneiss conductivities has been studied, but the relevant charge carriers have not been discussed in detail. Based on the various slopes of the linear relationship of logarithmic conductivities and reciprocal temperatures, it was proposed that the charge carriers in the gneiss sample at low and high temperatures are different. At high temperatures and pressures, the electrical properties of biotite-bearing felsic gneiss (with varied alkaline ionic contents) have been investigated [18]. It was proposed that chemical composition has a significant influence on the electrical conductivity of gneiss, and the electrical properties of gneiss markedly depend on the alkali ionic content [18]. Regarding the conduction mechanisms of gneiss, it was found to be impurity conduction with low activation enthalpies (33.60–55.68 kJ/mol) at low temperatures and ionic conduction with high activation enthalpies (68.16–83.52 kJ/mol) at high temperatures [18]. Under the conditions of 1.0 GPa and 300–890 K, the conductivities of granulite and temperatures conform to an Arrhenius relation above 500 K, whereas the behavior of electrical conductivity of the sample is unstable below 500 K [15]. The charge carriers of the granulite sample have not been studied. However, the conduction mechanism for granulite changes from an extrinsic one to an intrinsic one after three heating/cooling cycles. For dry eclogite, the electrical conductivities significantly increase with the increase of temperature and oxygen fugacity and slightly decrease with rising pressure. Small polarons were proposed to be the charge carriers of dry eclogite. The conductivities of dry eclogite cannot be used to explain the high conductivity layers (HCLs) under the stable middle–lower crust. Meantime, the high-conductivity anomalies under the Dabie–Sulu metamorphic belt were not caused by the presence of dry eclogite [17]. 

In the present study, the electrical conductivities of three natural schists with different contents of alkaline ions were measured under the experimental conditions. The influence of temperature, pressure, and chemical composition on the conductivities of schist was discussed in detail, and the microcosmic conduction mechanism of schist was investigated based on the activation enthalpies and previous relevant studies. Furthermore, we inferred the relationship between the electrical properties of schist and the high-conductivity anomalies under the Tibetan Plateau.

## 2. Experimental Methods

### 2.1. Sample Collection and Analysis

Three schists were collected from different rock masses in Xinjiang, China. The distribution of the main minerals in each schist sample was homogeneous, and the schist samples were completely fresh, unoxidized, and unfractured. The mineral compositions in the schist samples were detected by the scanning electron microscope (SEM) at the State Key Laboratory of Ore Deposit Geochemistry, Institute of Geochemistry, Chinese Academy of Sciences (CAS), Guiyang, China. The major element content of the schist samples was analyzed by X-ray fluorescence (XRF) by Australian Laboratory Services, Shanghai, China. The mineral compositions of schists are listed in Table 1; the samples were composed of feldspar, quartz, and biotite with various grain sizes. This indicated that our schist samples belong to biotite–plagioclase–quartz schist. The grain sizes of feldspar, quartz, and biotite in different schist samples were relatively similar to each other (Table 1). The major element contents of different samples were varied (Figure 1 and Table 2), and *C*_A_ for the three schist samples was 3.94, 5.17, and 5.78 wt.%, respectively. The schist samples were machined into regular cylinders (diameter: 6 mm; length: 6 mm) and then placed in a drying oven at the temperature of 473 K to keep them dry for subsequent in situ electrical conductivity measurements.

### 2.2. Impedance Measurements

The electrical conductivity in situ measurements on the schist samples at 0.5–2.5 GPa and 623–1073 K were carried out in the YJ-3000t multianvil apparatus and the Solartron-1260 impedance/gain-phase analyzer at the Key Laboratory of High-Temperature and High-Pressure Study of the Earth’s Interior, Institute of Geochemistry, Chinese Academy of Sciences, Guiyang, China. As shown in Figure 2, the sample assemblage for electrical conductivity measurements on schist was composed by a series of components. In order to avoid the influence of absorbed water on the electrical conductivity measurements, all components (pyrophyllite cube, ceramic tubes, MgO, and Al_2_O_3_ sleeves) were baked at 1073 K for 12 h in a muffle furnace. Pyrophyllite cube (32.5 × 32.5 × 32.5 mm^3^) and three-layer stainless steel sheets with a total thickness of 0.5 mm were used as the pressure medium and heater, respectively. MgO and Al_2_O_3_ sleeves adjacent to the heater were used to insulate and transmit pressure. A layer of nickel foil (thickness: 0.025 mm) was installed between the alumina and magnesia sleeves to shield against the external electromagnetic and spurious signal interference. The sample was placed into the MgO capsule, and the electrodes were composed of two nickel disks (diameter: 6.0 mm; thickness: 0.5 mm). In order to avoid the influence of free water on the electrical conductivity measurements, the experimental assembly was dried at 330 K in an oven before electrical conductivity measurements.

In the experiments, pressure and temperature were successively increased to the designated values. When the given temperature was stable, the impedance spectra of schist were collected in the frequency range of 10^−1^–10^6^ Hz. At a certain pressure, the temperature interval for electrical conductivity measurements is 50 K. In order to ensure the experimental data was reproducible, the electrical conductivities of the samples were measured in multiple heating/cooling cycles at 0.5–2.5 GPa (pressure error: ± 0.1 GPa) and 623–1073 K (temperature error: ± 5 K).

## 3. Results

Figure 3 shows the representative Nyquist diagram of the complex plane for the schist sample DS20 at 1.5 GPa and 623–1073 K. All complex impedance spectra of schist consist of two parts: an ideal semicircle in frequency region of 10^6^–10^3^ Hz and an additional tail in the frequency region of 10^3^–10^−1^ Hz (Figure 3). Previous studies have proposed that the ideal semicircle stands for the impedance and capacitive reactance of the whole sample and that the additional tail is caused by the diffusion effect of charge carriers between the sample and electrode [4,18,19,20]. In order to fit the electrical resistance of schist, we applied the equivalent circuit which is shown in Figure 3. It is of note that the equivalent circuit was analogous to those used in relevant previous studies, and it was explained in detail by Dai et al. [18]. For the resistance, the fitting errors were about 5%. The electrical conductivities of schist were obtained based on the following formula:(1)σ=L/SR
where *L* represents sample length (m), *S* represents the cross-sectional area of the electrodes (m^2^), *R* represents the sample resistance (Ω), and *σ* represents the electrical conductivity of the sample (S/m). 

The logarithmic electrical conductivities of DS20 in two heating/cooling cycles were plotted against the reciprocal temperatures at 1.5 GPa and 623–1073 K (Figure 4 and Table 3). As shown in Figure 4, the conductivities of DS20 measured at the same temperature in different cycles were close to each other. This indicated that the schist sample remained in a steady state during the entire heating/cooling cycles and our experimental data are reproducible. Under the conditions of a certain pressure and 623–1073 K, the electrical conductivities dramatically increased with the increasing temperature, and the conductivities and temperatures followed an Arrhenius relation. Effect of pressure on the conductivities of schist was much weaker than that of temperature, and the conductivities slightly increased with rising pressure (Figure 5 and Table 4). In addition, the electrical conductivities of schists markedly increased with the rising content of alkali ions (Na^+^ and K^+^) at low temperatures and became close to each other at high temperatures (Figure 6 and Table 5). Unexpectedly, there was no direct correlation between the conductivities of the schist samples and iron content (Figure 6 and Table 2). Furthermore, the conductivities of the schist samples did not change regularly with the feldspar content (Table 1 and Figure 6). Under the conditions of a certain pressure and 623–1073 K, electrical conductivities of schist and temperatures conformed to the Arrhenius relation; the relevant formula was explained by Dai et al. [2]. The thermodynamic parameters of the schists were fitted by the Arrhenius relation. Table 6 showed that the activation enthalpies of the three schists were 44.16–61.44 kJ/mol and were negatively correlated with alkali ionic content. The pre-exponential factors for the Arrhenius relation between the conductivities of the schist samples and temperatures are positive values.

## 4. Discussion

### 4.1. Comparison with Previous Studies

The electrical conductivities of schist showed good repeatability in the multiple heating/cooling cycles (Figure 4), revealing that the schist sample remained stable throughout the entire experimental process. In this process, the rock structure and matter distribution of schist samples were not obviously changed. For most silicate rocks, the conductivities in the first heating/cooling cycles were unstable, reflecting that the chemical compositions and porosity are gradually adjusted [16,17,18]. The discrepancy between schist and other silicate rocks may be due to their different mineralogical assemblages and structures.

The conductivities of the schist samples were slightly related to pressure. This again confirmed the weak dependence of the conductivities of silicate minerals and rocks on the pressure [21,22]. Under the experimental conditions, the electrical conductivities of the schist samples with different alkaline ionic contents were ~10^−3.5^–10^−1.5^ S/m. In order to research the influence factors of conductivity of schist in detail, it was significant to investigate the significance of the dominant minerals (quartz, feldspar, and biotite) for the electrical conductivity of the schist sample. Wang et al. [8] researched the conductivities of polycrystalline quartz under the conditions of 1.0 GPa and 850–1600 K. The conductivities of polycrystalline quartz were lower than those of the schist samples. This discrepancy was proposed to result from the different internal structures and charge carrier contents of synthetic quartz and natural schist. Hu et al. [23] explored the conductivities of alkali feldspar solid solutions at 1.0 GPa and 873–1173 K. This work revealed that electrical conductivities of albite were 0.5 orders of magnitude higher than those of K-feldspar and that the conductivities of solid solutions increased with Na^+^ content at constant temperature. The alkaline ions were proposed to be the charge carriers for the alkali feldspars. It was showed that the conductivities of schists are much higher than those of alkali feldspars [18]. It implies that the conductivities of the schist samples are not dominated by feldspars. Li et al. [9] has investigated the conductivities of phlogopite at 1.0 GPa and 473–1173 K. The conductivities of schist were lower than the values of phlogopite at higher temperatures and higher than those of phlogopite at lower temperatures. In addition, the influence of temperature on the conductivity of phlogopite was more significant than that of schist [9]. Furthermore, the electrical conductivities of some rocks with the similar mineralogical assemblage of the schist samples have been studied. Dai et al. [2] investigated the conductivities of natural granite samples (main minerals: quartz, feldspar, and biotite) under the thermodynamic conditions of the Earth’s middle–lower crust. However, the conductivities of schists were much higher than the values of granites (Figure 7). At high temperatures and pressures, the electrical conductivity of gneiss was investigated in detail [18]. Figure 7 showed that the conductivities of schists are much higher than those of gneisses. Fuji-ta et al. [15] investigated the electrical conductivities of granulite under middle-lower crustal pressure and temperature conditions. The conductivities of the schist samples were close to those of granulite at low temperature and higher than those of granulite at high temperature. In addition, the influence of temperature on the conductivities of schist was more significant than that of granulite. These discrepancies may be due to the different chemical compositions and rock structures of schist and other relevant rocks.

### 4.2. Conduction Mechanism

For the schist samples, the electrical conductivities and temperatures conformed to an Arrhenius relation at 0.5–2.5 GPa and 623–1073 K. This implied that one conduction mechanism dominates the conductivities of schist samples at a certain pressure. As a very complicated natural metamorphic rock, the conduction mechanism for schist was difficult to determine. Biotite is a hydrous mineral with a high content of iron, and feldspar contains abundant alkali ions. Therefore, possible conduction mechanisms for the schist samples include hydrogen-related defect conduction, small polaron conduction, intrinsic ionic conduction, and impurity ionic conduction. The conductivities of schists did not change consistently with the content of calcium and alkali ions (Table 2 and Figure 6), and the Δ*H* for the schist samples were smaller than those for minerals (feldspars and phlogopite) and rocks (granite and gneiss) displaying intrinsic ionic conduction [2,9,10,18,23]. This indicated that intrinsic ionic conduction was not the conduction mechanism for the schist samples. Furthermore, hydrogen-related defect and small polaron conduction mechanisms have been proposed to be the dominant ones for many iron-bearing hydrous minerals and rocks [3,24,25,26,27,28,29,30]. Δ*H* for iron-bearing hydrous minerals and rocks with hydrogen-related defect conduction is about 57.60–76.80 kJ/mol [3,25,29] and about 52.80–76.80 kJ/mol for iron-bearing hydrous minerals and rocks with small polaron conduction [26,28,30]. For the schist samples, biotite is the sole iron-bearing or hydrous mineral. Although the activation enthalpies for electrical conductivity of schist were close to those for hydrogen-related defect conduction and small polaron conduction of iron-bearing hydrous minerals and rocks, the conductivities of schist did not regularly change with biotite content increase. Based on the electron backscattered images (Figure 1), the biotite is uniformly distributed in schist samples to a certain extent. Interconnectivity of biotite in DS20 was better than that in DS18 and DS19, but electrical conductivity of DS20 was lower than that of DS18 and DS19. Meanwhile, there was no relationship between the electrical conductivity of schist and iron content (Table 2 and Figure 6). It was implied that the charge carriers for the schist samples were not protons or small polarons. Finally, impurity K^+^ and Na^+^ ions are widely distributed in many minerals and rocks [2,8,18]. The activation enthalpies for the electrical conductivity of schist (44.16–61.44 kJ/mol) are very close to the values of gneiss with impurity ionic conduction (33.60–55.68 kJ/mol). Meantime, the electrical conductivities of schist increased with the increase of *C*_A_ (*C*_A_ = K_2_O + Na_2_O = 3.94, 5.17, and 5.78 wt.%) at low temperatures. In the high-temperature region, the conductivities of the schist samples were close to each other. It was caused by the influence of activation enthalpies, which decreased with the increasing content of alkali ions. Furthermore, Δ*H* increased or remained roughly constant upon pressure (Table 6), which was the evidence for ionic conductivity. Therefore, the conduction mechanism for the schist samples was proposed to be impurity ionic conduction (charge carriers: K^+^ and Na^+^).

## 5. Geophysical Implications

The schist samples were collected in Altai, Xinjiang, China. The Chinese Altai, as part of the peri-Siberian orogenic system, was amalgamated with the Kazakhstan/south Mongolian orogenic system in the latest Paleozoic. A range of metamorphic rocks (up to amphibolite faces) were widely distributed in the Chinese Altai, and an important rock in this setting is schist with the main mineral constituents being quartz, feldspar, and biotite [31,32]. The Tibetan plateau is another globally significant orogenic belt, and schist with the main minerals being quartz, feldspar, and biotite is a crucial metamorphic rock in this region [33]. Therefore, it is valuable to study the relationship between biotite–feldspar–quartz schist and electrical structure in the Tibetan Plateau. The Tibetan Plateau, a consequence of the ongoing India–Eurasia continental collision since approximately 50–60 Ma, has been widely investigated over the past decade [34,35,36]. The deep structure, evolution, and tectonic processes under the Tibetan Plateau have been comprehensively studied by magnetotelluric (MT) results [36,37,38]. However, the material compositions and thermodynamic state under the Tibetan Plateau are not very clear. It is necessary to combine filed magnetotelluric (MT) results and laboratory-derived electrical conductivities of natural rocks to infer the internal states of the Tibetan Plateau. Schist with various chemical compositions is widely distributed in the Tibetan Plateau, where it coexists with other regional metamorphic rocks [12,13,14,39,40]. The interior of the Tibetan Plateau must distribute a mass of metamorphic rocks with different degrees of metamorphism. According to the magnetotelluric data, high-conductivity anomalies are widely distributed in the Earth’s middle–lower crust and upper mantle [36,41]. The interpretations for the high-conductivity anomalies are discrepant [1,3,29,42,43,44,45], and the high-conductivity layers in various regions may be constituted by different materials. Therefore, it is crucial to systematically study the causes for high-conductivity anomalies under the Tibetan Plateau. The conductivity–depth profile of schist under the Tibetan Plateau was obtained based on the conductivity–temperature data of the sample and the relationship between temperature and depth in the Earth’s stationary crust [46]:(2)$$T=T0+(\frac{Q}{k})Z-(\frac{A_{0}}{2k})Z^{2}$$
where *T*_0_ is the surface temperature (K), *Q* is the surface heat flow (mW/m^2^), *Z* is the lithosphere layer depth (km), *k* is thermal conductivity (W/mK), and *A*_0_ is the lithospheric radiogenic heat productivity (μW/m^3^). According to previous studies, the corresponding thermal calculation parameters for the Tibetan Plateau are *Q* = 84 mW/m^2^ [47], *A*_0_ = 0.8 μW/m^3^ [48], *T*_0_ = 280.5 K, and *k* = 2.6 W/mK [49].

According to the heat conduction equation (Equation (2)) and the conductivity–temperature results of schists, we constructed the conductivity–depth profiles of schists beneath the Tibetan Plateau. The two conductivity–depth profiles were constructed for HCLs that moderately correlate with the northern and southern Tibetan Plateau, respectively (Figure 8). As shown in Figure 8, the electrical conductivities of high-conductivity layers at the depth of 5–35 km in the Tibetan Plateau are about 10^−3^–1.0 S/m. The electrical conductivities of schists are lower than the largest conductivities of the high-conductivity layers in Tibetan Plateau. It is indicated that the presence of schist with main minerals of quartz, feldspar, and biotite cannot cause the high-conductivity anomalies in the middle to lower crust beneath the Tibetan Plateau. For anomalously high electrical conductivities, previous studies have confirmed that interconnected aqueous fluids and melts are important factors [3,42]. According to the MT results and geological structural setting, it was suggested that crustal melt beyond the Kunlun Fault permeated into the northern Tibetan Plateau [38], and significant amounts of aqueous fluids overlying a layer of partial melting constitute the HCLs in the southern Tibetan Plateau [37,50]. Therefore, the high-conductivity layers beneath Tibetan Plateau might be caused by the presence of interconnected aqueous fluids and melts.

## 6. Conclusions

Electrical conductivity of schist was found to significantly depend on temperature, following an Arrhenius relation at high temperatures and pressures. Effect of pressure on the conductivity of schist was much weaker than that of temperature. Chemical composition had an important influence on the conductivities of the schist samples. Electrical conductivities of schists increased markedly with increasing content of alkali ions (K^+^ and Na^+^) at low temperature and were close to each other at high temperature. According to the activation enthalpies (44.16–61.44 kJ/mol) and previous studies, alkaline ions (K^+^ and Na^+^) were proposed to be the charge carriers at high temperatures and pressures. Furthermore, the high-conductivity layers beneath Tibetan Plateau are not composed of schist with main minerals of quartz, feldspar, and biotite.

## Figures and Tables

**Figure 1 materials-12-00961-f001:**
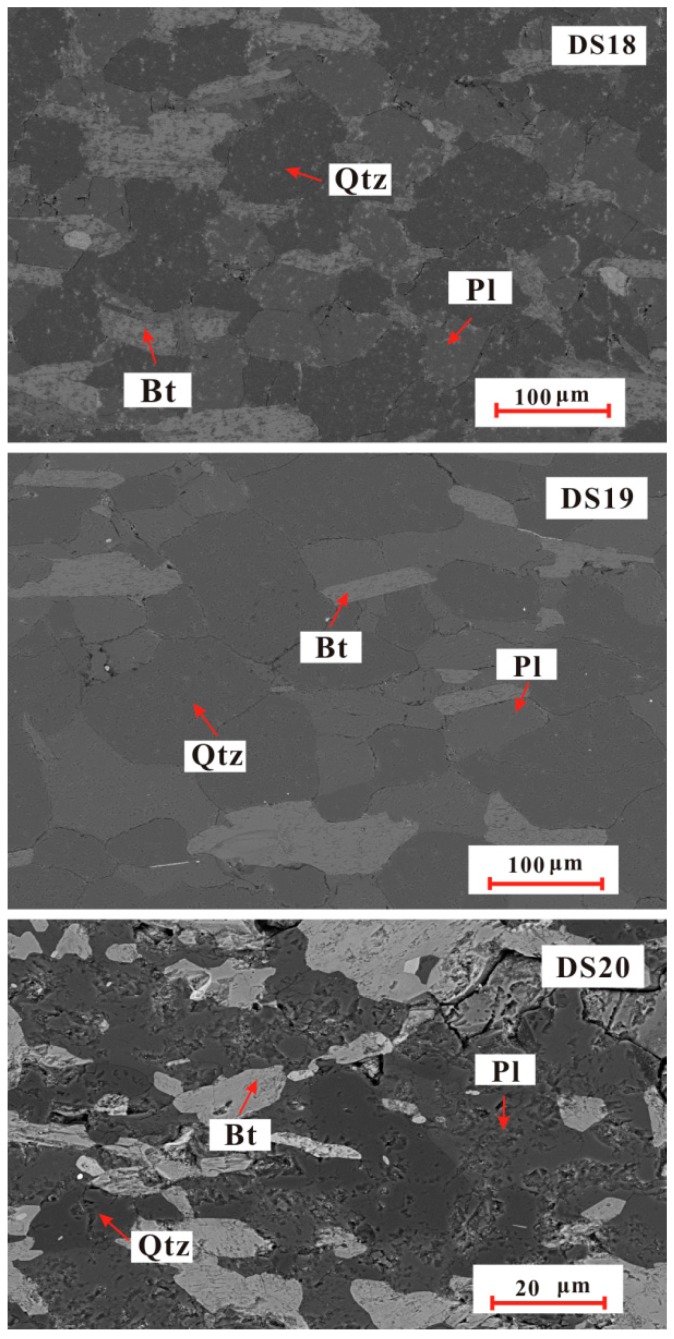
Scanning electron microscope (SEM) images of the schist samples. Pl, Qtz, and Bt represent plagioclase, quartz, and biotite, respectively.

**Figure 2 materials-12-00961-f002:**
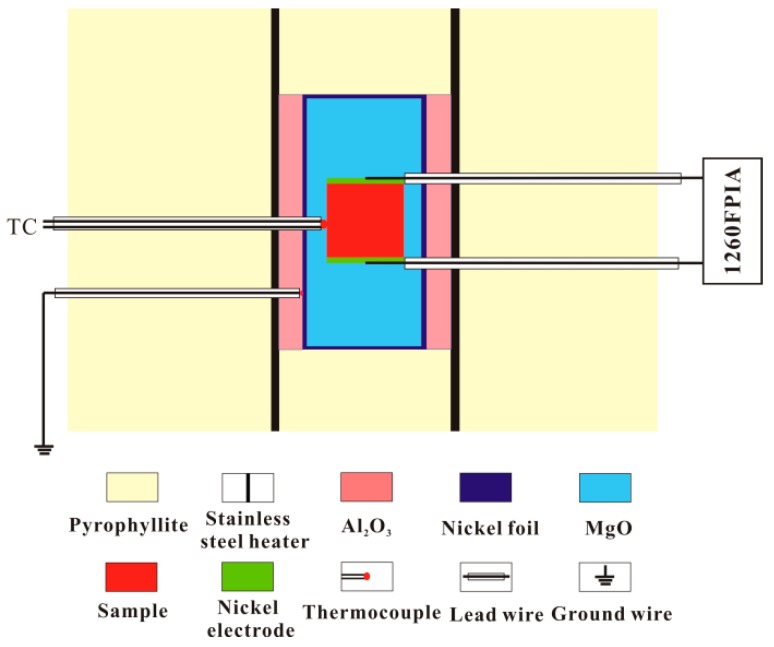
Schematic drawing of the sample assembly for the present experiments. Note: TC stands for the thermocouple.

**Figure 3 materials-12-00961-f003:**
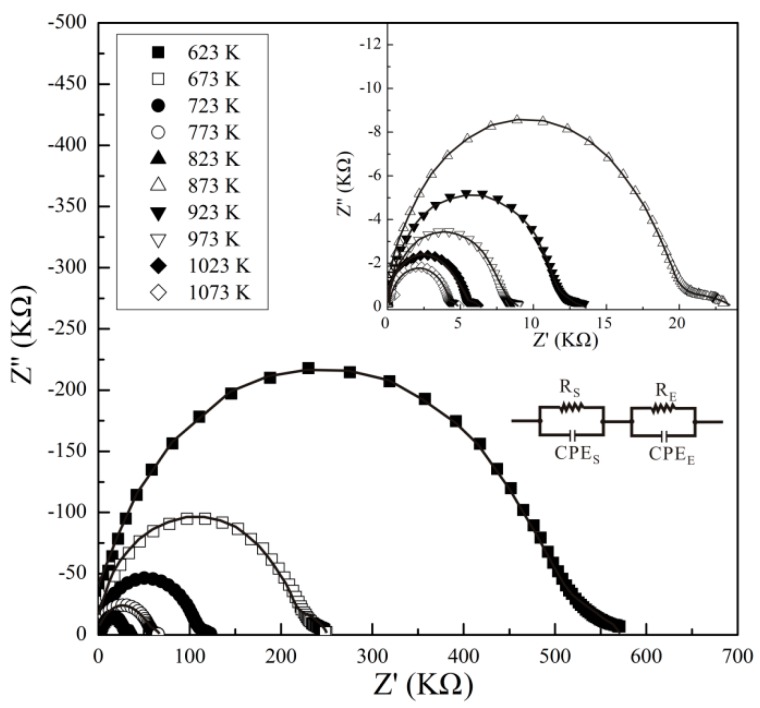
Nyquist plot of the complex impedance for sample DS20 at 1.5 GPa and 623–1073 K.

**Figure 4 materials-12-00961-f004:**
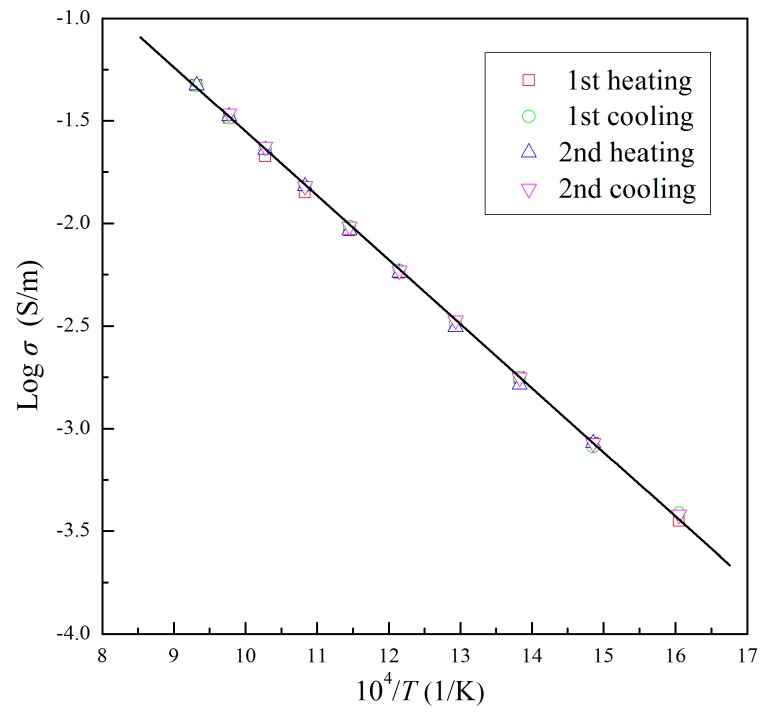
Logarithm of electrical conductivity versus reciprocal temperature for DS20 during two heating/cooling cycles at 1.5 GPa.

**Figure 5 materials-12-00961-f005:**
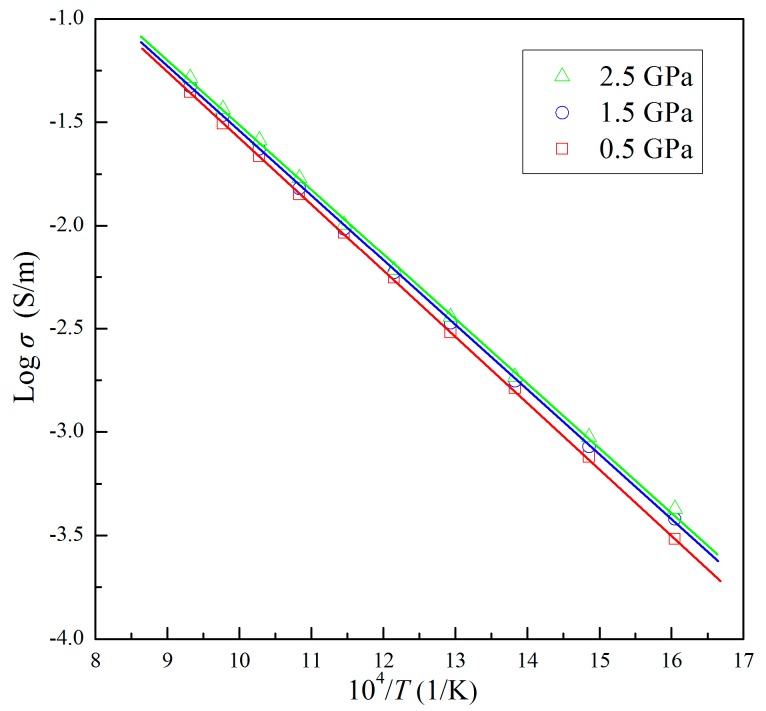
Logarithm of electrical conductivity versus reciprocal temperature for DS20.

**Figure 6 materials-12-00961-f006:**
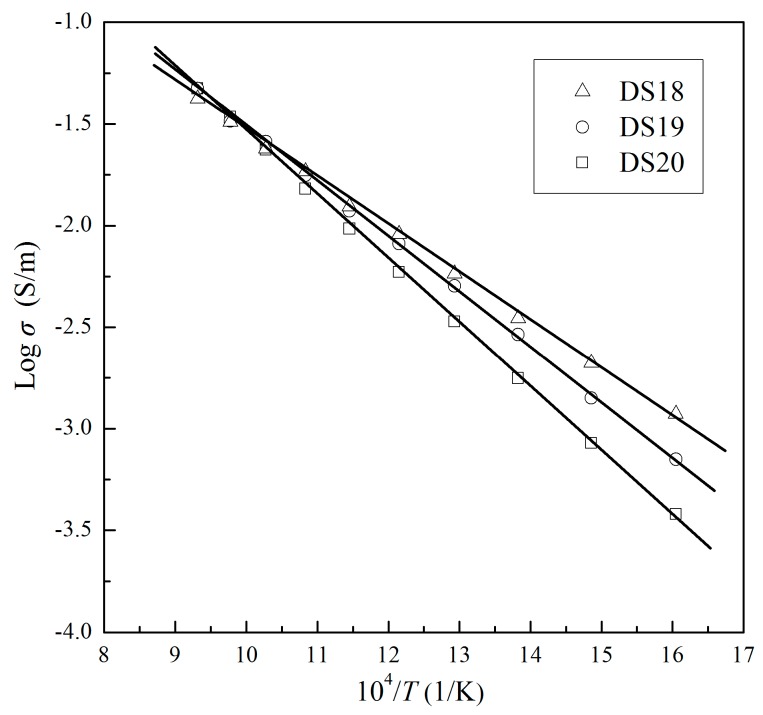
The relationship between electrical conductivity and temperature for the three schist samples.

**Figure 7 materials-12-00961-f007:**
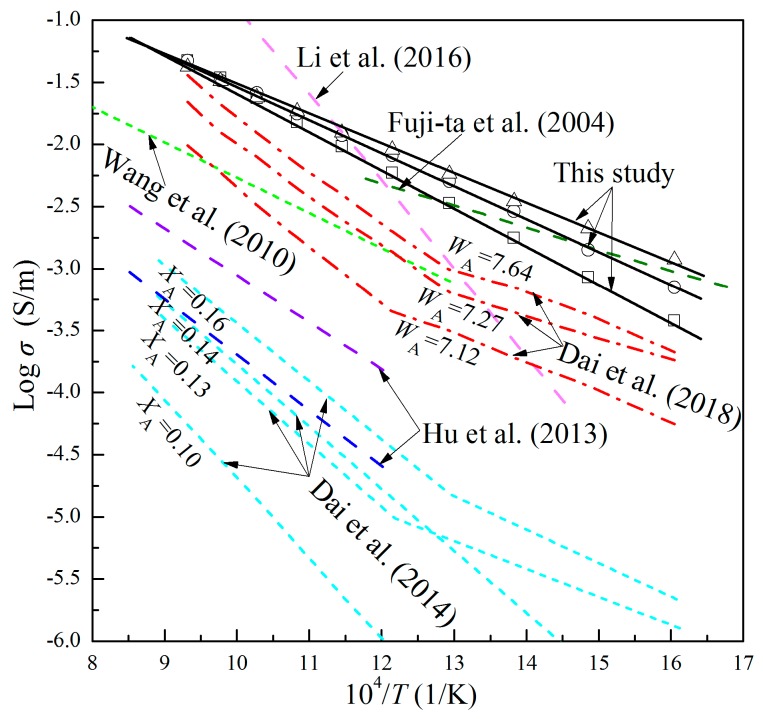
The electrical conductivities of schists and other relevant minerals and rocks at high temperatures and pressures. The dashed red lines represent the conductivities of gneisses with various chemical compositions (*W*_A_ = K_2_O + Na_2_O + CaO in weight percent) at 1.5 GPa [18]; the dashed sky-blue lines stand for the conductivities of granites with various chemical compositions (*X*_A_= (K_2_O + Na_2_O + CaO)/SiO_2_ in weight percent) at 0.5 GPa [2]; the dashed dark-green line represents the conductivities of granulite [15], the dashed purple and blue lines stand for the conductivities of albite and K-feldspar [23], respectively; the dashed pink line represents the conductivities of phlogopite [9]; and the dashed light-green line represents the conductivities of quartz [8]. The electrical conductivities of granulite, albite, K-feldspar, phlogopite, and quartz were measured at 1.0 GPa [8,9,15,23].

**Figure 8 materials-12-00961-f008:**
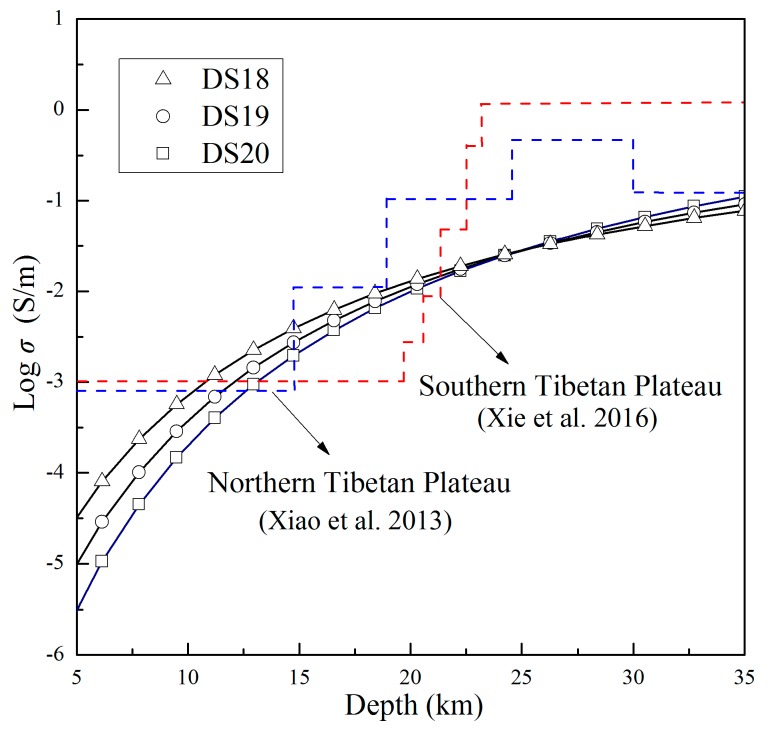
The conductivity–depth profiles for schists and electrical structures of high-conductivity layers beneath the Tibetan Plateau [36,41]. The solid black lines stand for the laboratory-based conductivity–depth profiles from data obtained for the schist samples in this study based on the thermal structure in the Tibetan Plateau, and the blue and red dashed lines represent the electrical structures of the northern Tibetan Plateau and southern Tibetan Plateau, respectively.

**Table 1 materials-12-00961-t001:** Mineral compositions of the schist samples. Pl, Qtz, and Bt represent plagioclase, quartz, and biotite, respectively.

Sample No.	Mineralogical Assemblage (in Volume Percent, Grain Size)
DS18	Pl (35%, 20 × 10 ‒ 80 × 60 μm^2^) + Qz (55%, 20 × 20 ‒ 80 × 80 μm^2^) + Bi (10%, 10 × 5 ‒ 100 × 20 μm^2^)
DS19	Pl (40%, 20 × 10 ‒ 110 × 80 μm^2^) + Qz (50%, 20 × 20 ‒ 100 × 100 μm^2^) + Bi (5%, 10 × 5 ‒ 120 × 20 μm^2^)
DS20	Pl (50%, 20 × 10 ‒ 60 × 40 μm^2^) + Qz (30%, 10 × 10 ‒ 60 × 60 μm^2^) + Bi (20%, 10 × 5 ‒ 50 × 10 μm^2^)

**Table 2 materials-12-00961-t002:** Chemical composition of the schist samples (the results of X-ray fluorescence analyses).

Oxides (wt.%)	DS18	DS19	DS20
SiO_2_	60.53	64.79	47.99
Al_2_O_3_	17.44	15.62	13.50
MgO	3.76	3.18	4.50
CaO	2.62	3.25	13.35
Na_2_O	2.94	3.00	3.76
K_2_O	2.84	2.17	0.18
Fe_2_O_3_	7.25	5.89	10.36
TiO_2_	0.77	0.84	2.10
Cr_2_O_3_	0.02	0.02	0.02
MnO	0.09	0.11	0.19
BaO	0.04	0.03	0.02
SrO	0.04	0.04	0.03
P_2_O_5_	0.17	0.23	0.26
SO_3_	<0.01	<0.01	<0.01
L.O.I	1.60	1.15	4.21
Total	100.11	100.33	100.47

Note: L.O.I denotes the loss on ignition.

**Table 3 materials-12-00961-t003:** The electrical conductivities of the schist sample DS20 over multiple heating/cooling cycles at 1.5 GPa and 623–1073 K.

*T* (K)	*σ* (×10^−3^ S/m)			
	1st Heating	1st Cooling	2nd Heating	2nd Cooling
623	0.35	0.38	/	0.38
673	0.85	0.82	0.85	0.85
723	1.77	1.77	1.63	1.77
773	3.26	3.26	3.12	3.37
823	5.74	5.89	5.74	5.89
873	9.23	9.65	9.23	9.65
923	14.15	15.16	15.16	15.16
973	21.22	23.07	22.82	23.58
1023	32.65	32.65	33.16	34.23
1073	47.16	/	47.16	/

Note: “/” represents the electrical conductivity (**σ**) is same as the value at the same temperature in the previous heating or cooling cycle.

**Table 4 materials-12-00961-t004:** Electrical conductivities of the schist sample DS20.

*T* (K)	*σ* (×10^−3^ S/m)		
	0. 5 GPa	1.5 GPa	2.5 GPa
623	0.30	0.38	0.42
673	0.76	0.85	0.94
723	1.63	1.77	1.85
773	3.03	3.37	3.60
823	5.58	5.89	6.06
873	9.23	9.65	10.11
923	14.15	15.16	16.98
973	21.65	23.58	25.88
1023	31.21	34.23	36.59
1073	44.21	47.16	51.76

**Table 5 materials-12-00961-t005:** The electrical conductivities of the schist samples DS18, DS19, and DS20 at 1.5 GPa and 623–1073 K.

*T* (K)	*σ* (×10^−3^ S/m)		
	DS18	DS19	DS20
623	1.18	0.71	0.38
673	2.11	1.41	0.85
723	3.49	2.91	1.77
773	5.79	5.05	3.37
823	9.07	8.16	5.89
873	12.40	11.79	9.65
923	18.47	17.68	15.16
973	23.83	25.88	23.58
1023	32.34	32.65	34.23
1073	42.10	47.16	47.16

**Table 6 materials-12-00961-t006:** The thermodynamic parameters for the electrical conductivities of the three schists.

Sample No.	*P* (GPa)	*T* (K)	Log *σ*_0_ (S/m)	Δ*H* (kJ/mol)	*R* ^2^
DS18	1.5	623–1073	0.77 ± 0.03	44.16 ± 0.95	99.91
DS19	1.5	623–1073	1.17 ± 0.03	51.84 ± 0.94	99.92
DS20	0.5	623–1073	1.62 ± 0.01	61.44 ± 0.86	99.99
	1.5	623–1073	1.58 ± 0.02	59.52 ± 0.90	99.97
	2.5	623–1073	1.63 ± 0.04	59.52 ± 0.96	99.90

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
