# Peer review of "Effect of Temperature, Pressure, and Chemical Composition on the Electrical Conductivity of Schist: Implications for Electrical Structures under the Tibetan Plateau"

_materials, 2019, doi:10.3390/ma12060961_

Round 1
Reviewer 1 Report
The work entitled "Effect of Temperature, Pressure and Chemical Compositions on the Electrical Conductivity of Schist: Implications for Electrical Structures under the Tibetan Plateau" by Wenqing Sun et. al., is very interesting one that connects the material properties with the geophysical structure in a complex tectonic area. It is written with a clear way and presents significant scientific interest.
It could be published in "Materials" after introducing some minor revision. Figs 5, 6 and 7 are the core of the work. The impact of the work will be increased if the authors include three tables with the data σ, P and T used for figures 6, 6 and 7 creation. This must be done before publication.
Author Response
Response to Reviewer 1 comments
The work entitled "Effect of Temperature, Pressure and Chemical Compositions on the Electrical Conductivity of Schist: Implications for Electrical Structures under the Tibetan Plateau" by Wenqing Sun et. al., is very interesting one that connects the material properties with the geophysical structure in a complex tectonic area. It is written with a clear way and presents significant scientific interest. It could be published in "Materials" after introducing some minor revision.
Thank the anonymous reviewer for very constructive and enlightened comments and suggestions in the reviewing process, which helped us greatly in improving the manuscript. According to the reviewer’s significant comments and suggestions, we have conscientiously revised our manuscript.
1. Figs 5, 6 and 7 are the core of the work. The impact of the work will be increased if the authors include three tables with the data σ, P and T used for figures 6, 6 and 7 creation. This must be done before publication.
Thanks for the anonymous reviewer’s valuable and important comments and suggestions. we have added three tables with the data σ, P and T used for figures 5, 6 and 7 creation. We believed that the three additional tables can clearly present our relevant experimental results, and significantly enhance the impact of this research. As shown below, Table 3, 4 and 5 correspond to Figure 5, 6 and 7, respectively.
Table 3. The electrical conductivities of the schist sample DS20 during two heating/cooling cycles at 1.5 GPa and 623‒1073 K.
T (K)
| σ (×103 S/m) | |||
1st heating | 1st cooling | 2nd heating | 2nd cooling | |
623 | 0.35 | 0.38 | / | 0.38 |
673 | 0.85 | 0.82 | 0.85 | 0.85 |
723 | 1.77 | 1.77 | 1.63 | 1.77 |
773 | 3.26 | 3.26 | 3.12 | 3.37 |
823 | 5.74 | 5.89 | 5.74 | 5.89 |
873 | 9.23 | 9.65 | 9.23 | 9.65 |
923 | 14.15 | 15.16 | 15.16 | 15.16 |
973 | 21.22 | 23.07 | 22.82 | 23.58 |
1023 | 32.65 | 32.65 | 33.16 | 34.23 |
1073 | 47.16 | / | 47.16 | / |
Table 4. The electrical conductivities of the schist sample DS20 under the conditions of 0.5‒1.5 GPa and 623‒1073 K.
T (K) | σ (×103 S/m) | ||
0. 5 GPa | 1.5 GPa | 2.5 GPa | |
623 | 0.30 | 0.38 | 0.42 |
673 | 0.76 | 0.85 | 0.94 |
723 | 1.63 | 1.77 | 1.85 |
773 | 3.03 | 3.37 | 3.60 |
823 | 5.58 | 5.89 | 6.06 |
873 | 9.23 | 9.65 | 10.11 |
923 | 14.15 | 15.16 | 16.98 |
973 | 21.65 | 23.58 | 25.88 |
1023 | 31.21 | 34.23 | 36.59 |
1073 | 44.21 | 47.16 | 51.76 |
Table 5. The electrical conductivities of the schist sample DS18, DS19 and DS20 at 1.5 GPa and 623‒1073 K.
T (K) | σ (×103 S/m) | ||
DS18 | DS19 | DS20 | |
623 | 1.18 | 0.71 | 0.38 |
673 | 2.11 | 1.41 | 0.85 |
723 | 3.49 | 2.91 | 1.77 |
773 | 5.79 | 5.05 | 3.37 |
823 | 9.07 | 8.16 | 5.89 |
873 | 12.40 | 11.79 | 9.65 |
923 | 18.47 | 17.68 | 15.16 |
973 | 23.83 | 25.88 | 23.58 |
1023 | 32.34 | 32.65 | 34.23 |
1073 | 42.10 | 47.16 | 47.16 |

Reviewer 2 Report
Multi-anvil systems excert quasi-hydrostatic pressure to the specimen. Can the authors estimate the divergence of the true pressure from being hydrostatic?
The authors extract the dc conductivity from the dielectric data (see Fig. 3). Then, Fig. 4 is useless and space consuming to appear.
According to Table3, ΔH increases or remains roughly constant upon pressure. This evidences about ionic conductivity (such as Na+, K+ transport).
Author Response
Response to Reviewer 2 comments
1. Multi-anvil systems excert quasi-hydrostatic pressure to the specimen. Can the authors estimate the divergence of the true pressure from being hydrostatic?
Thanks for your very important and professional comments. It’s true that multi-anvil apparatus can only provide quasi-hydrostatic pressure to the specimen. Generally, the differential stress of the sample capsule is determined by the sample size and materials of components of sample assembly. According to plenty of experimental studies using the same sample assembly (Hu et al. 2014; Hui et al. 2015; Dai et al. 2016), the divergence of the true pressure from being hydrostatic is about ± 0.1 GPa. We have added the errors of temperature and pressure (± 5 K and ± 0.1 GPa) in our revised manuscript.
2. The authors extract the dc conductivity from the dielectric data (see Fig. 3). Then, Fig. 4 is useless and space consuming to appear.
Thanks for your very important and professional comments. According to the valuable suggestion, we have deleted Fig. 4 in our revised manuscript.
3. According to Table3, ΔH increases or remains roughly constant upon pressure. This evidences about ionic conductivity (such as Na+, K+ transport).
Thanks for your very significant and professional comments. You provided a new and crucial evidence for that the charge carriers of the schist sample are alkaline ions (Na+ and K+). The activation enthalpies for the electrical conductivity of schist (44.16–61.44 KJ/mol) are very close to the values of gneiss with impurity ionic conduction (33.60–55.68 KJ/mol), and smaller than those for minerals (feldspars and phlogopite) and rocks (granite and gneiss) displaying intrinsic ionic conduction (Hu et al. 2011, 2014; Dai et al. 2014, 2018). Meantime, the electrical conductivities of schist increased with the increasing content of alkali ions (K+ and Na+) at low temperatures. In the high temperature region, the conductivities of the schist samples were close to each other. It was caused by the influence of activation enthalpies which decreased with the increasing content of alkali ions. Therefore, the conduction mechanism for the schist samples was proposed to be impurity ionic conduction (charge carriers: K+ and Na+).
References
Dai, L.D., Hu, H.Y., Li, H.P., Wu, L., Hui, K.S., Jiang, J.J. and Sun, W.Q. (2016) Influence of temperature, pressure, and oxygen fugacity on the electrical conductivity of dry eclogite, and geophysical implications. Geochemistry, Geophysics, Geosystems, 17, 2394‒2407.
Dai, L.D., Hu, H.Y., Li, H.P., Jiang, J.J., Hui, K.S. (2014) Influence of temperature, pressure, and chemical composition on the electrical conductivity of granite. American Mineralogist, 99, 1420‒1428.
Dai, L.D., Sun, W.Q., Li, H.P., Hu, H.Y., Wu, L., Jiang, J.J. (2018) Effect of chemical composition on the electrical conductivity of gneiss at high temperatures and pressures. Solid Earth, 9, 233‒245.
Hu, H.Y., Dai, L.D., Li, H.P., Jiang, J.J. and Hui, K.S. (2014) Electrical conductivity of K-feldspar at high temperature and high pressure. Mineralogy and Petrology, 108, 609–618.
Hu, H.Y., Li, H.P., Dai, L.D., Shan, S.M., Zhu, C.M. (2011) Electrical conductivity of albite at high temperatures and high pressures. American Mineralogist, 96, 1821–1827.
Hui, K.S., Zhang, H., Li, H.P., Dai, L.D., Hu, H.Y., Jiang, J.J. and Sun, W.Q. (2015) Experimental study on the electrical conductivity of quartz andesite at high temperature and high pressure: evidence of grain boundary transport. Solid Earth, 6, 1037–1043.
